# Intra-Abdominal Hypertension: A Systemic Complication of Severe Acute Pancreatitis

**DOI:** 10.3390/medicina58060785

**Published:** 2022-06-10

**Authors:** Carla Mancilla Asencio, Zoltán Berger Fleiszig

**Affiliations:** Critical Care Unit and Gastroenterology Section, Department of Internal Medicine, Clinical Hospital, University of Chile, Santiago 8380456, Chile; carlitamancilla@uchile.cl

**Keywords:** abdominal pressure, intraabdominal hypertension, abdominal compartment syndrome

## Abstract

Patients with severe acute pancreatitis (SAP) present complications and organ failure, which require treatment in critical care units. These extrapancreatic complications determine the clinical outcome of the disease. Intra-abdominal hypertension (IAH) deteriorates the prognosis of SAP. In this paper, relevant recent literature was reviewed, as well as the authors’ own experiences, concerning the clinical importance of IAH and its treatment in SAP. The principal observations confirmed that IAH is a frequent consequence of SAP but is practically absent in mild disease. Common manifestations of AP such as pain, abdominal distension, and paralytic ileus contribute to increased abdominal pressure, as well as fluid loss in third space and aggressive fluid replacement therapy. A severe increase in IAP can evolve to abdominal compartment syndrome and new onset organ failure. Conservative measures are useful, but invasive interventions are necessary in several cases. Percutaneous drainage of major collections is preferred when possible, but open decompressive laparotomy is the final possibility in some cases in order to definitively reduce abdominal pressure. Intra-abdominal pressure should be measured in all SAP cases that worsen despite adequate treatment in critical care units. Conservative measures must be introduced to treat IAH, including negative fluid balance, digestive decompression by gastric–rectal tube, and prokinetics, including neostigmine. In the case of insufficient responses to these measures, minimally invasive interventions should be preferred.

## 1. Background

When talking about intra-abdominal hypertension (IAH), we must consider acute pancreatitis (AP) as a systemic disease and IAH as an associated organ dysfunction. Comprehension of the evolution of AP has evolved in the last decades, becoming clear that prognosis is closely related to organ failure rather than local compromise. Since the Atlanta symposium in 1992 [1], questioning has emerged on the relevance of local complications and some isolated systemic manifestations. A retrospective analysis covering 10 years in the Mayo Clinic showed that local complications and single transient organ failure were related to morbidity, while persistent organ failure was related to mortality [2]. The Atlanta Revision 2013 distinguished three categories (mild, moderately severe, and severe AP), defining severe pancreatitis as the entity complicated with persistent (more than 48 h) organ failure, considering respiratory, cardiovascular, and renal dysfunction [3]. According to the modified Marshall score, a patient is to be considered in organ failure if they present with at least one of the following: moderate hypoxemia manifested in (PaO2/FiO2) a ratio <300, systolic blood pressure <90 mmHg not responsive to fluids, or an elevation of creatinine >1.9 mg/dL [4].

IAH is a process in which a progressive increase in intra-abdominal pressure (IAP), not compensated by abdominal cavity compliance, results in multiple physiological consequences, including renal dysfunction, splanchnic hypoperfusion, a decrease in cardiac output, and a reduction in respiratory system compliance. Abdominal compartment syndrome (ACS) is the most serious form of IAH, characterized by elevated abdominal pressure associated with organ failure and a high mortality if not resolved [5]. In critical care units, IAH has been described in association with various medical and surgical conditions, with the following being the most important: abdominal trauma, major burns, aortic aneurysm surgery, liver transplantation, severe septic shock, massive fluid overload, and severe AP (SAP). IAH is a risk factor capable of doubling mortality in critical care patients [6]. In AP, IAH has also been associated with a detriment in prognosis. As IAH evolves to cardiovascular, renal, and respiratory dysfunction, and, in accordance with Atlanta 2013 and the modified Marshall score, it can be suggested that IAH underlies some cases of SAP.

## 2. Magnitude of the Problem

Although overestimated in the initial reports, IAH is a frequent complication in SAP. Current consensus by the World Society of Abdominal Compartment Syndrome (WSACS), updated in 2013, defines IAH as a persistent elevation of IAP over 12 mmHg, and ACS is defined as an IAP over 20 mmHg, associated with a new organ failure/dysfunction [7] (Table 1).

Considering these criteria, the incidence of IAH in SAP is about 60% and 10–30% for ACS in patients admitted to an intensive care unit [8,9]. In patients with mild forms of AP, IAH is almost nonexistent [10].

The impact of IAH on the evolution of AP is negative. Even though most data come from retrospective cases series, the results are consistent: AP patients with IAH/ACS more frequently present with organ dysfunction (manifested in severity scores as SOFA or APACHE II), pancreatic necrosis, persistent systemic inflammatory response (SIRS), need for invasive interventions, feeding intolerance, longer hospitalization, and mortality [11,12]. IAH is an early manifestation in the evolution of SAP occurring most of the time in the first days of the illness. When present, it is almost always in association with or preceding other systemic complications and tends to normalize in patients who improve. On the contrary, persistent IAH and particularly ACS is a factor predictive of mortality [12,13]. Mortality related to ACS has been described as being between 30 to 60%. In a group of 12 patients who required decompressive laparotomy, despite initial improvement, mortality reached 50% [14,15]. A recent prospective international observational study conducted by Marcos-Neira et al. analyzed a cohort of 301 patients with SAP admitted to intensive care with at least one organ failure: 91% had IAH and 34% ACS. Grade IV IAH was associated with infected necrosis, need for surgery, invasive mechanical ventilation, and requirement of renal replacement therapy. The mortality was 28% [16].

## 3. Why Is AP Prone to Develop IAH? Risk Factors

IAH results from an imbalance between the abdominal cavity contents and compliance to support them. The abdomen is a closed compartment with a limited capacity to expand towards the anterior wall and diaphragmatic face. IAP increases in parallel to the volume increase, but, after reaching a critical point, a smaller increase in volume will result in an exponential increment in pressure [5]. Factors that influence IAP and pressure/volume relation can be categorized the following way:Factors that decrease abdominal wall compliance;Factors that increase abdominal contents;Factors that increase capillary leak/fluid overload.

Table 2 summarizes the factors that may increase IAP in AP.

Fluid balance is a crucial risk factor for IAH/ACS. In a prospective cohort of AP patients, those who received more than 4.1 L of fluids in 24 h presented more organ failure, fluid collections, and respiratory and renal insufficiency [17]. A meta-analysis of randomized control trials in 2886 AP patients showed that aggressive fluid resuscitation (defined as 3–5 mL/kg/h in the first 24 h) was associated with a relative risk of more than two times for acute kidney injury and pulmonary edema. Although IAP was not measured, IAH may underlie both respiratory and renal failure in this scenario [18]. A randomized control trial in 76 AP patients compared a group reanimated to an infusion rate of 10–15 mL/kg/h versus 5–10 mL/kg/h until hemodynamic stability. The rapid infusion group received significantly more fluids and evolved with a higher APACHE II score, rate of mechanical ventilation, ACS (72 versus 32%), and significantly higher mortality [19].

IAH is a marker of severity in AP, generally accompanied by other systemic manifestations. In a prospective cohort in a tertiary care center, Dambrauskas et al. showed that severity scores of APACHE II >7, Glasgow-Imrie >3, and MODS score >2 at admission were predictive of ACS [10]. Ke et al. found 24 h fluid balance, number of collections, and lower calcium levels were risk factors for IAH, and Bezmarevic et al. found a correlation between procalcitonin level and IAP level at 24 h of admission, which were both significantly higher in SAP patients and non survivors [20,21].

Obesity has been recognized as a predictor of worst prognosis in AP. Excessive adiposity comprising both parenchyma and peripancreatic fat is prone to lipolysis, inflammation, and necrosis. Fat tissue produces proinflammatory cytokines involved in systemic manifestations. On the other hand, obesity is a mechanical risk factor for IAH. However, it should be noted that obese patients develop compensatory mechanisms due to a chronic elevated IAP, so moderate levels of IAH must be carefully evaluated before deciding therapeutic options [22].

Regarding images, Gupta et al. compared imaging findings on computed tomography in patients with and without IAH. The presence of collections, maximum diameter, and volume of collection, and the presence of moderate pleural effusion were risk factors for IAH [23]. The study by Verma et al. comparing the CT findings in a group of moderate and SAP with and without IAH showed that the IAH group more frequently presented pancreatic necrosis >50%, moderate-gross ascites, and the “round belly sign”. The last consisted of an increase of >0.8 for the relation of anteroposterior/and transverse diameter of the abdomen measured at the level of the left-renal vein crossing aorta, excluding the subcutaneous fat. Although these imaging findings have not been consistently related to IAH/ACS in SAP, they deserve attention as warning signs that should prompt IAP measurement [24].

## 4. Pathophysiology of IAH in AP

IAH can produce damage in AP through systemic and local mechanisms. The splanchnic vascular territory is the most susceptible to IAP increase. IAH can further compromise pancreatic perfusion, contributing to pancreatic necrosis. Intestinal hypoperfusion may lead to bacterial translocation resulting in a second insult, subsequent organ dysfunction, and eventually early mortality. Bacterial translocation may also be implied in pancreatic necrosis infection [25]. In patients with SAP and abdominal sepsis Al-Bahrani et al. found a correlation between elevated IAP, high levels of procalcitonin, and low levels of anti-endotoxin immunoglobulins (a marker of gut barrier dysfunction), which normalized with the resolution of IAH [26]. In animal models, IAH has been shown to produce pancreatic histological damage similar to acute pancreatitis, including leukocyte infiltration, mitochondrial damage, and necrosis [27]. Figure 1 describes pathways to organ dysfunction associated with IAH [28].

## 5. Treatment of IAH/ACS in AP

Therapeutic options in IAH/ACS are varied and can be categorized according to the underlying mechanism increasing IAP. These interventions are not exclusive, as there are often several factors involved. In the same way, as IAH is a process, interventions can play a prophylactic or therapeutic role and may be sequentially applied. All of these strategies are suggested in the treatment of IAH/ACS, independent of the cause, and it should be noted that most of them are not based on high quality trials, so the strength of recommendation in WSACS consensus is sometimes weak or absent [29]. With respect to sedation and analgesia, there are no trials demonstrating their effect on IAP. Nonetheless, WSACS maintained the recommendation to ensure adequate anxiety and pain relief in IAH. Analgesia is also a cornerstone topic in the management of a painful disease such as AP. Although not studied in AP, the bolus administration of cisatracurium was shown to lower IAP in patients with IAH due to fluid overload, abdominal trauma, and burns. Brief trials of neuromuscular blocking agents are actually considered safe and useful in a variety of situations in the critically ill, including IAH [30,31].

Table 3 details medical and surgical treatment options.

Our group communicated at The European Pancreas Club Meeting 2009, an experience in 18 patients with SAP admitted to intensive care. The mean APACHE II score was 14.6, and 13 patients (72%) developed IAH. In this subset, three patients evolved with ACS. All of the patients received medical treatment to lower their IAP. Therapies included promotility agents, nasogastric suction, sedation, analgesia, and vasopressors to assure APP > 60 mmHg. Eleven patients (61%) underwent percutaneous drainage of the fluid collection. One patient with ACS and anuria (IAP 33 mmHg) required decompressive laparotomy, with full recovery. Lesser sac was not explored nor was any debridement done. There was no mortality in this group [32].

We will review specific interventions studied in the setting of AP.

### 5.1. Therapies to Evacuate Intraluminal Contents

A recent RCT by He et al. evaluated the efficacy of neostigmine to lower IAP in AP with IAH. Neostigmine was administered at a dose of 1 mg intramuscular every 12 h, which was eventually increased to 1 mg every 8 or 6 h in the case of no defecation. The rate of decrease in IAP was significantly faster in the neostigmine group. Neostigmine had a rapid onset of action, lowering IAP 3 h after the first dose. IAP was reduced by approximately 20% in 24 h. The stool volume was higher in the neostigmine group during the 7-day observational period. No adverse events related to neostigmine were observed [33].

A study using Chinese medicine by Zhang et al., who randomized 80 patients with SAP, deserves mention. Here, 40 patients received a herbal decoction (Da Cheng Qi) in the form of enema combined with the external use of Glaubert salt (derived from sodium sulfate), once a day for seven days. Both medications have laxative properties. The treated group experienced earlier relief of pain and, by the fourth day, a significant decrease in IAP (from 14 to 6 mmHg) and APACHE II score. Hospitalization time was 16 versus 29 days in the control group, with no differences in mortality [34].

### 5.2. Therapies to Evacuate Intra-Abdominal Collections

Sun et al. randomized 110 patients with SAP (APACHE II score 17) to receive conservative treatment versus an indwelling abdominal catheter placed on day one, with the aim to drain abdominal fluids and measure IAP. By the third day, in the treatment group, the APACHE II score and IAP decreased significantly in parallel with the drainage of 2900 mL of fluid from the abdominal cavity. The relief of abdominal pain and hospitalization time were also shorter and the rate of cyst formation was lower [35].

In order to compare percutaneous catheter drainage versus peritoneal lavage, He et al. conducted a randomized control trial in 86 patients with SAP. The underlying theory was that pancreatic ascites fluid is rich in toxins and proinflammatory cytokines. No differences were found in terms of mortality and organ dysfunction, but the percutaneous drainage group experienced a significant drop in IAP at 24, 48, and 72 h, while the peritoneal lavage group persisted with an elevated IAP [36].

### 5.3. Therapies to Restore Fluid Balance and Capillary Leak

There are some experiences suggesting that hemodialysis/hemofiltration act through a dual mechanism involving negative fluid balance and the removal of inflammatory mediators. Oda et al., in 17 patients with AP treated with continuous hemodiafiltration, found a decrease in IAP and the levels of IL-6 [37]. In the study by Xu et al., 25 patients with SAP and ACS were submitted to continuous veno-venous hemofiltration (CVVHF). After 24 h, IAP decreased from 22.9 to 17.2 mmHg, accompanied by a decrease in the levels of TNFα. The survival rate was 92%. The authors suggested that the removal of humoral mediators was the factor improving prognosis, but no details were given with respect to fluid balance [38]. In a retrospective analysis by Pupelis et al. on 75 AP patients receiving versus 55 not receiving CVVHF, treatment resulted in a negative fluid balance, lower infection rates (particularly pneumonia), and shorter hospital stay, without changes in mortality [39]. In a more recent trial, Xie et al. randomized 74 AP patients to receive CVVHF from one to seven days. Patients with IAP < 20 mmHg had no beneficial effects. Patients with IAP > 20 mmHg, who received the therapy, had a shorter ICU stay, less days on mechanical ventilation, and lower 28-day operation rates. This group also had more days with a negative fluid balance, suggesting a benefit from this mechanism. There were no differences in mortality [40].

### 5.4. Abdominal Decompression

As occurs in general ACS, the optimal threshold for surgical decompression in ACS associated with AP is not clear. Refractory ACS and ACS resistant to medical therapy seem a reasonable indication, but many concerns persist about timing, IAP cut level, and risks. Boone et al. described a group of 12 patients undergoing decompressive laparotomy due to AP/ACS. All patients had been previously submitted to multiple medical interventions. The mean APACHE II was 23 and the median time to diagnosis of ACS was 4.5 days. Decompression was performed before 7 days in nine patients. Pancreatic and peripancreatic debridement was performed in six patients. After 24 h of laparotomy, multiple variables experienced improvement: (PaO2/FiO2) ratio, airway pressure, urinary output, and APACHE II. Despite this, six patients died (50% mortality). Mortality was related to age [15]. Mentula et al. described 26 AP/ACS patients. The mean SOFA score was 12 and the mean IAP was 31.5 mmHg. Decompression resulted in renal and respiratory improvement. Overall mortality was 46% and decreased to 18% in patients operated on in the first 4 days [41]. In a matched case control study by Husu et al., 47 patients with SAP were submitted to open abdomen management. In 40/47 patients (85%), the indication for laparotomy was refractory ACS and in 68% the urinary output was less than 20 mL/h. Mortality was 42% in the open abdomen group and 34% presented visceral ischemia, significantly higher than the matched control group [42]. In a retrospective cohort study, Kao et al. analyzed a subgroup of 47 patients with necrotizing pancreatitis and refractory ACS. The APACHE II score was 22 and the severity index in computed tomography was 8 or greater. Patients were severely ill, as follows: 100% had sepsis, 72% had septic shock, 85% were on mechanical ventilation, 76% had renal failure in renal replacement therapy, and 68% had hepatic dysfunction. All of the patients underwent decompressive laparotomy, necrosectomy, multiple drainages, ileostomy, and feeding jejunostomy. The mean interval from admission to surgery was 25 days. The overall complication rate was 31% (mainly local bleeding) and, of note, mortality was only 8.5% [43].

Despite these data, two different situations must be recognized. Early decompressive laparotomy (before 7 days) with intention to treat ACS should avoid any exploration of lesser sac and necrosectomy, which can increase morbidity and mortality. In cases of late ACS associated with infected necrosis, necrosectomy can be considered in association with decompressive laparotomy in cases not amenable to resolve through minimally invasive approaches.

## 6. Prevention

The role of enteral nutrition was evaluated in a trial by Sun et al. Enteral nutrition increases blood flow of the gut mucosa, stimulates the intestinal motility, preserves gut integrity, and prevents bacterial-endotoxin translocation, thus decreasing infectious complications. However, the effect of enteral nutrition on IAP and feeding tolerance in IAH remains uncertain. In a pilot prospective study, 60 patients with SAP defined by Atlanta 1992 and admitted to intensive care were randomly assigned to receive early enteral nutrition (24 h after admission) via a nasojejunal tube versus delayed nutrition (8th day after admission). The delayed group received total parenteral nutrition the first week. By the 9th day, the incidence of IAH was higher in the delayed nutrition group. During the first 3 days, abdominal distension was more frequent in the early nutrition group. IAP < 15 was associated with less feeding intolerance. On the other hand, early enteral nutrition did not increase IAP [44].

The idea of reaching hemodynamic stability at the expense of as little fluid as possible is based on previously discussed observations, showing that high amounts of volume increase the risk of IAH, respiratory, and renal failure. The addition of colloid hydroxyethyl starch to normal saline or Ringer’s lactate for initial hydration in AP has been shown to reduce the amount of volume administered and the risk of IAH [45,46]. However, the use of colloids in critical care has been discouraged due to increased risk of renal failure and mortality in septic patients [47]. Current guidelines in AP suggest the use of balanced crystalloids for initial resuscitation [48,49].

## 7. Discussion

IAH is a well-documented phenomenon with multiple risk factors in accordance with the physiology of abdominal cavity and systemic consequences. There is enough evidence supporting that IAH has a pathogenic role in SAP and is not an epiphenomenon. Despite this, the topic has not received the same attention compared with other topics in SAP. AP presents general and local conditions that explain the high frequency of IAH in severe cases. Influence in prognosis is consistent with an increase in morbidity, including renal, respiratory, and cardiovascular dysfunction and mortality, particularly related to ACS. IAH may also be implied in local complications such as necrosis and infection. Measuring IAP is a low cost–low risk maneuver. The technique is widely known and standardized. Cost–benefit relation supports the measurement of IAP in SAP. The focus should be on moderate to severely ill patients in the early phase of the disease. Prompt detection of IAH is a marker of prognosis and permits initiating therapies in order to prevent progression to grade III–IV IAH and ACS. These authors suggest a low threshold of suspicion to measure IAP, considering it in patients with any of the following factors: obesity, clinical evidence of early organ dysfunction (low urinary output, increased work for breathing, and high requirements of fluids to maintain hemodynamic stability), presence of SIRS, ileus or feeding intolerance, presence of radiological alarm signs as extensive pancreatic necrosis/fluid collections, and moderate to severe pleural effusion/ascites. IAP recording is mandatory in patients with SAP (persistent organ failure) and those admitted to intensive care. The potential benefit of detecting a patient with IAH overcomes the costs of measuring IAP in terms of risks and supplies. Moreover, published data show that low-grade IAH tends to be a less serious complication, unlike high grade III–IV IAH and ACS, reinforcing the idea that the early detection and management of IAH warrants an easier treatment and better prognosis. Regarding prevention, maybe the most useful measure is the cautious use of fluids. This is not in contradiction with the benefit of an early and targeted resuscitation in patients who need it.

It should be highlighted that most patients with criteria of IAH and even ACS are able to respond to non-operative strategies, which must be encouraged in all of the cases. On the other hand, refractory ACS should prompt surgical decompression due to elevated mortality when delayed. This decision will be always difficult and should be held by a multidisciplinary team, taking into consideration that natural evolution of ACS is almost always fatal. Advances in critical care, rationalization of surgery indication, early enteral nutrition, judicious use of fluids, and the emerging role of minimally invasive procedures have positively impacted on the prognosis of AP, and general morbidity and mortality have diminished.

## 8. Conclusions

IAH is practically absent in mild acute pancreatitis (MAP), but it is frequently present in SAP. IAH is not only a factor that can be useful in the prediction of worse prognosis, but a phenomenon that can determine an ominous clinical outcome. Thus, the surveillance of IAH in the context of SAP should be considered a routine intervention and efforts should be made to initiate early treatment in order to prevent further increase and progress to ACS.

## Figures and Tables

**Figure 1 medicina-58-00785-f001:**
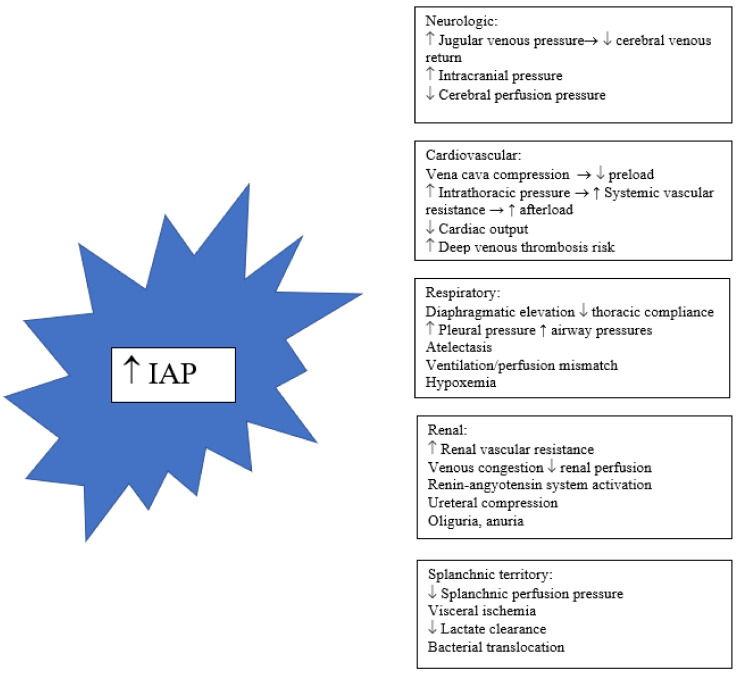
Systemic effects of intra-abdominal hypertension.

**Table 1 medicina-58-00785-t001:** Consensus definitions by the World Society of Abdominal Compartment Syndrome (WSACS).

Normal IAP	0 to 5–7 mmHg
IAH	Persistent elevation of IAP > 12 mmHg: Grade I: 12–15 mmHg Grade II: 16–20 mmHg Grade III: 21–25 mmHg Grade IV: >25 mmHg
ACS	IAP > 20 mmHg + new organ failure (SOFA score ≥ 3)
APP	PAM-IAP
Measuring technique	Patient in complete supine position Introduce 25 mL of sterile saline into the empty bladder Transducer zeroed at the level of the midaxillary line After 60 s at end expiration, IAP is registered in a monitor IAP should be expressed in mmHg

IAP: intra-abdominal pressure IAH: intra-abdominal hypertension, ACS: abdominal compartment syndrome APP: abdominal perfusion pressure, PAM: mean arterial pressure SOFA: Sequential Organ Failure Assessment.

**Table 2 medicina-58-00785-t002:** General and local factors increasing IAP in AP.

Mechanism	Eventually Present in AP
Factors that decrease abdominal wall compliance	Pain Abdominal wall edema Obesity
Factors that increase abdominal contents	Gastroparesis Ileus Ascites Retroperitoneal hemorrhage-edema Acute fluid collections
Factors that increase capillary leak/fluid overload	Massive fluid overload SIRS/MODS

SIRS: Systemic inflammatory response syndrome. MODS: Multiorgan dysfunction syndrome.

**Table 3 medicina-58-00785-t003:** Medical and surgical treatment of IAH/ACS.

Therapies to improve abdominal wall compliance	Sedation and analgesia Neuromuscular blockade Consider supine position <20° Avoid prone position Remove constrictive dressings and abdominal eschars
Therapies to evacuate intraluminal contents or intra-abdominal collections	Nasogastric/colonic decompression Promotility agents, laxatives Decompressive colonoscopy Discontinuation of enteral nutrition Percutaneous drainage Paracentesis
Therapies to restore fluid balance	Restriction of fluids Negative fluid balance Use of diuretics/albumin Hemodialysis/ultrafiltration
Therapies to maintain APP >60 mmHg	Fluids/vasoactive drugs
Abdominal decompression	Decompressive laparotomy Alba line fasciotomy

## Data Availability

Not applicable.

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
