# Peer review of "Intra-Abdominal Hypertension: A Systemic Complication of Severe Acute Pancreatitis"

_medicina, 2022, doi:10.3390/medicina58060785_

Round 1

Reviewer 1 Report

This review is very well written (comprehensive and informative) and I commend the authors. May be, the authors can detail the literature available on predictive factors of intra-abdominal hypertension in severe acute pancreatitis (Verma S, Rana SS, Kang M, Gorsi U, Gupta R. Computed tomography features predictive of intra-abdominal hypertension in acute necrotizing pancreatitis: A prospective study. Indian Journal of Gastroenterology. 2021 Jun;40(3):326-32.)

Author Response

Dear Sir,

We are grateful for your comments. The suggested article (Verma S, Rana SS, Kang M, Gorsi U, Gupta R. Computed tomography features predictive of intra-abdominal hypertension in acute necrotizing pancreatitis: A prospective study. Indian Journal of Gastroenterology. 2021 Jun;40(3):326-32), was included and analyzed in the section of risk factors.

Reviewer 2 Report

This manuscript summarizes current knowledge considering intra-abdominal hypertension in patients with severe acute pancreatitis. Authors focus on pathophysiology of intra-abdominal hypertension, specificity of intra-abdominal hypertension in case of acute pancreatitis and principles of therapy. In this paper, relevant recent literature was reviewed. Authors emphasize the fact that intra-abdominal hypertension is common systemic complication in severe acute pancreatitis associated with organ failure and worse outcome. Therefore, surveillance of intra-abdominal hypertension should be considered in routine management of patients with severe acute pancreatitis. Early recognition of intra-abdominal hypertension and adequate therapy may prevent progress to abdominal compartment syndrome.

I have a few comments considering this manuscript.

1. In the page No. 2, there is sentence…..Considering this criteria, incidence of IAH in AP is about 60%.......  it is wrong….there should be emphasize that this high incidence of IAH (60%) is in severe AP….

2.  In this paper, authors discuss the treatment options including therapies to evacuate intraluminal contents, therapies to restore fluid balance and surgical therapy based on abdominal decompression. In my opinion, authors should mention also treatment options how to improve abdominal wall compliance in more details (sedation and analgesia, neuromuscular blockade etc.).

3. Considering abdominal decompression and decompressive laparotomy. It is very important to differentiate early intra-abdominal hypertension with early decompressive laparotomy (< 7 days) and late intra-abdominal hypertension with late laparotomy. There are two different situations. Early decompressive laparotomy is used just for abdominal decompression only and necrosectomy of pancreatic necrotic tissue is not recommended due to high risk of complication and absence of infection pancreatic necrosis in this early period. On the opposite side, late intra-abdominal hypertension is usually associated with infected pancreatic necrosis a laparotomy should release intra-abdominal pressure and surgeon has to perform necrosectomy as well. Authors should mention this fact in manuscript.    

Author Response

Dear Sir,

We are grateful for Your suggestions and comments, which have been considered in the new manuscript. References have been adapted as well.

  • The reviewer is right, we added “S” to AP, this proportion is only true in severe acute pancreatitis (SAP). In our original text we wanted to affirm this proportion concerning AP patients admitted to critical care units, i.e. SAP, but this correction made the affirmation clearer.

  • We added a paragraph analyzing the limited available information on sedation, analgesia and muscular blockade in the treatment of intra-abdominal hypertension:

“With respect to sedation and analgesia there are no trials demonstrating their effect on IAP. Nonetheless WASCS maintained the recommendation to ensure adequate anxiety and pain relief in IAH. Analgesia is also a cornerstone topic in management of a painful disease as AP. Although not studied in AP, bolus administration of cisatracurium showed to lower IAP in patients with IAH due to fluid overload, abdominal trauma and burns. Brief trials of neuromuscular blocking agents are actually considered safe and useful in a variety of situations in critically ill, including IAH [30,31]”.

  • The reviewer is right again and points a fact that was relevant to clarify. We added the next paragraph:

“Despite these data two different situations must be recognized. Early decompressive laparotomy (before 7 days) with intention to treat ACS should avoid any exploration of lesser sac and necrosectomy, which can increase morbidity and mortality. In cases of late ACS associated with infected necrosis, necrosectomy can be considered in association with decompressive laparotomy in cases not amenable to resolve by minimally invasive approaches”.

 We hope that these changes are satisfactory and contribute to ameliorate the quality of our paper.